# Historical Data-Based Prescribed Performance Optimal Control for PMSM Systems with Disturbance

1st Bowen Zhang
*College of Control Science and Engineering*
*Bohai University*
Jinzhou, China
2022008014@qymail.bhu.edu.cn

*Abstract*—This paper proposed a prescribed performance optimal control method for permanent magnet synchronous motor (PMSM) systems with nonlinear disturbance. Firstly, a novel prescribed performance approach is proposed to constrain both the state and error of PMSM systems in order to obtain high precision control performance. Then, an adaptive dynamic programming method based on historical data is designed to solve the optimal controller. Furthermore, an identifier based on neural network is constructed to approximate the system disturbances. Finally, a simulation example is implemented to validate the efficacy of the proposed control method.

*Index Terms*—Permanent Magnet Synchronous Motor, Prescribed Performance Control, Adaptive Dynamic Programming.

## I. Introduction

Permanent Magnet Synchronous Motors (PMSM) are widely applied in robotics and industrial fields due to their advantages such as higher power density, higher torque-inertia ratio, and higher efficiency [1]. In the current context of energy shortages, how to effectively enhance the working efficiency of PMSM and thereby conserve more energy has emerged as a primary concern. Optimal control theory is often utilized to conserve control resources by minimizing performance index functions. [2] et al. propose a nonlinear optimal controller and observer schemes based on a $\theta$-D approximation approach for PMSM systems, which can effectively elevate the operational efficiency of PMSM control systems. However, existing methods need repeated checks of the adaptive updating law to ensure the satisfaction of the persistent excitation (PE) condition. [3] et al. introduced an adaptive dynamic programming approach based on historical data to obtain the optimal solution for the system, effectively eliminating the need for PE signals. Currently, designing an optimal control method for PMSM systems that can dispense with the PE condition poses a challenging task.

Existing optimal control strategies for PMSM systems often struggle to meet high-precision control requirements. Prescribed performance control represents an effective approach to achieving high-performance control demands by constraining system performance. [4] et al. proposed a prescribed performance optimal control method to enhance control performance. Existing prescribed performance methods typically impose constraints only on system errors. [5] et al. introduced a low-complexity prescribed performance approach that can simultaneously constrain both system states and errors. This method is equally applicable to PMSM systems. However, this method lacks discussion of system disturbance issues. Therefore, our current primary research motivation is to develop an anti-disturbance prescribed performance method for implementing optimal control of PMSM systems.

Taking inspiration from the above, this paper proposed a prescribed performance optimal control method for permanent magnet synchronous motor (PMSM) systems with nolinear disturbance. The main contributions are listed below:

1) A novel prescribed performance approach is proposed to constrain both the state and error of PMSM systems in order to obtain high precision control performance. System state and system error can be constrained simultaneously.
2) An adaptive dynamic programming method based on historical data is designed to solve the optimal controller. The PE conditions required by most adaptive dynamic programming methods can be eliminated.
3) An identifier based on neural network is constructed to approximate the system disturbances.

## II. Preliminaries

### A. System Description

In the field-oriented control, the PMSM has the following physical model in the d-q frame:

$$\begin{cases} \dot{\omega} = -\frac{\mathbf{B}}{\mathbf{J}}\omega - \frac{\mathbf{T}_L}{\mathbf{J}} + \frac{1.5 n_p \psi_F}{\mathbf{J}} + \frac{D_\tau}{\mathbf{J}} \\ \dot{i_d} = -\frac{\mathbf{R}_S}{\mathbf{L}_d}i_d + n_p\omega i_q + \frac{1}{\mathbf{L}_d}u_d + D_d \\ \dot{i_q} = -n_p\omega i_d - \frac{\mathbf{R}_S}{\mathbf{L}_q}i_d - \frac{n_p\psi_F}{\mathbf{L}_q}\omega + \frac{1}{\mathbf{L}_q}u_q + D_q \end{cases} \tag{1}$$

where $\omega$, $n_p$, $\mathbf{T}_L$ and $\mathbf{J}$ are the angular velocity, the number of pole pairs, load torque, and rotor inertia. $i_q$, $i_d$, $u_q$ and $u_d$ denote the stator current and voltage of q-axis and d-axis, respectively. $\mathbf{L}_d$ is the stator inductance and $\mathbf{R}_S$ is the

stator resistance. $\mathbf{B}$ and $\psi_F$ stand for the viscous frictional coefficient and the rotor flux linkage. The model (1) can be rewritten as

$$\dot{\mathbf{X}}(t) = f(\mathbf{X}(t)) + gu(t) + d(t), \tag{2}$$

where $\mathbf{X} = [\omega, i_d, i_q]^T$ is the system state,

$$f(\mathbf{X}(t)) = \begin{bmatrix} -\frac{\mathbf{B}}{\mathbf{J}}\omega - \frac{\mathbf{T}_L}{\mathbf{J}} + \frac{1.5 n_p \psi_F}{\mathbf{J}} \\ -\frac{\mathbf{R}_S}{\mathbf{L}_d} i_d + n_p \omega i_q \\ -n_p \omega i_d - \frac{\mathbf{R}_S}{\mathbf{L}_q} i_d - \frac{n_p \psi_F}{\mathbf{L}_q}\omega \end{bmatrix}, \quad g = \begin{bmatrix} 0 & 0 \\ \frac{1}{\mathbf{L}_q} & 0 \\ 0 & \frac{1}{\mathbf{L}_d} \end{bmatrix},$$

$u(t) = [u_d, u_q]^T$ represents the voltage of q-axis and d-axis, $d(t) = [\frac{D_\tau}{\mathbf{J}}, D_d, D_q]^T$ is the system disturbance.

### B. NN identifier

A NN identifier is introduced to estimate the PMSM system with disturbance, with the aim of mitigating the negative impacts of disturbances on the system.

The function $d(t)$ can be approximated by the NN in the following:

$$d(t) = \mathbf{W}_{\mathbf{i}}^{*T}\sigma_{\mathbf{i}} + \epsilon_{\mathbf{i}}, \tag{3}$$

where $\mathbf{W}_{\mathbf{i}}^* \in R^{n_{ef}*3}$ is ideal weight matrix, $\sigma_{\mathbf{i}} \in R^{n_{ef}}$ is the NNs activation function, $\epsilon_{\mathbf{i}}$ is bounded approximation error, $n_{ef}$ represents the number of neurons in the NNs.

Let $\hat{\mathbf{W}}_{\mathbf{i}}$ denote the estimation, the adaptive identifier is built as

$$\dot{\hat{\mathbf{X}}}(t) = -k_a\tilde{\mathbf{X}}(t) + f(\mathbf{X}(t)) + gu(t) + \hat{\mathbf{W}}_{\mathbf{i}}^T\sigma_{\mathbf{i}}, \tag{4}$$

where $\hat{\mathbf{X}}(t)$ is the identifier state, $\tilde{\mathbf{X}}(t) = \hat{\mathbf{X}}(t) - \mathbf{X}(t)$ is the identification error, $k_a$ is a positive parameter, and $\tilde{\mathbf{W}}_{\mathbf{i}}$ is the estimation error of $\hat{\mathbf{W}}_{\mathbf{i}}$.

The updating law of $\hat{\mathbf{W}}_{\mathbf{i}}$ is designed as

$$\dot{\hat{\mathbf{W}}}_{\mathbf{i}} = -\mathbb{T}\sigma_{\mathbf{i}}\tilde{\mathbf{X}}(t)^T - k_b\mathbb{T}\hat{\mathbf{W}}_{\mathbf{i}} \tag{5}$$

where $\mathbb{T} \in R^{n_{ef}*n_{ef}}$ is the positive definite gain matrix and $k_b$ is the positive design parameter.

### C. System Constraint

This section considers the situation where both the tracking error and all system states are constrained simultaneously. Before a discussion, the tracking error of PMSM is defined as

$$e_t(t) = \mathbf{X}(t) - \mathbf{X}_d(t), \tag{6}$$

where $\mathbf{X}_d(t)$ is the reference signal.

The proposed control scheme needs to satisfy simultaneous constraints on both the tracking error and all system states of the PMSM system, namely, meeting the following conditions:

- Prescribed performance constraints:

$$-P_i(t) < e_{ti}(t) < P_i(t), \quad i = 1, 2, 3. \tag{7}$$

- Full-state constraints:

$$-\alpha_{ci} < \mathbf{X}_i(t) < \alpha_{ci}, \quad i = 1, 2, 3. \tag{8}$$

where $\mathbf{X}_1(t)$, $\mathbf{X}_2(t)$ and $\mathbf{X}_3(t)$ represent the states $\omega$, $i_d$, and $i_q$ of the PMSM system, respectively, and $\alpha_{ci}$ indicates the maximum value that the system states can withstand.

### D. Error Transformation

To implement the performance constraints in the previous section, the prescribed performance function is defined as

$$P_i(t) = ln(\frac{1 + \varpi_i(t)}{\varpi_i(t) - 1}), \tag{9}$$

where $\varpi_i(t)$ is the shifting function, is chosen as $\varpi_i(t) = \begin{cases} (1 - \imath_i)(\frac{T_{ci}-t}{T_{ci}})^{n+2} + \imath_i, & t \in [0, T_{ci}), \\ \imath_i, & t \in [T_{ci}, +\infty), \end{cases}$ with $\imath_i = \frac{\exp(\frac{\alpha_{ci}}{2\alpha_{bi}}) + \exp(-\frac{\alpha_{ci}}{2\alpha_{bi}})}{\exp(\frac{\alpha_{ci}}{2\alpha_{bi}})) - \exp(-\frac{\alpha_{ci}}{2\alpha_{bi}}))}$ where $\alpha_{ci}$ and $\alpha_{bi}$ is a positive design constant.

To achieve the prescribed performance (7), a mapping error transformation is introduced as

$$\xi_i(t) = \frac{\exp(\frac{e_{ci}}{2}) - \exp(-\frac{e_{ci}}{2})}{\exp(\frac{e_{ci}}{2}) + \exp(-\frac{e_{ci}}{2})}, \quad e_{ci} = \frac{e_{ti}}{\alpha_{ci}}. \tag{10}$$

Then, we introduce an error transformation related to $\xi_i$

$$\eta_i(t) = \varpi_i(t)\xi_i(t), \quad \eta_i(0) = 0. \tag{11}$$

where $\eta_i(t)$ is another transformation error.

In order to guarantee $\eta_i(t) \in (-1, 1)$ for all $t \geq 0$. An error transformation is given as

$$\varepsilon_i(t) = \frac{\eta_i(t)}{1 - \eta_i(t)^2}, \quad \varepsilon_i(0) = 0. \tag{12}$$

Through a series of error transformations (9)-(12), the tracking problem of the original constrained PMSM system is transformed into a stabilization one of the following unconstrained system:

$$\dot{\varepsilon}(t) = \mathcal{A}_a(\dot{\mathbf{X}}(t) - \dot{\mathbf{X}}_d(t)) - \mathcal{A}_b \tag{13}$$

where

$$\mathcal{A}_a = \begin{bmatrix} \frac{\mu_1\alpha_{c1}\nu_1}{\omega_1^2} & 0 & 0 \\ 0 & \frac{\mu_2\alpha_{c2}\nu_2}{\omega_2^2} & 0 \\ 0 & 0 & \frac{\mu_3\alpha_{c3}\nu_3}{\omega_3^2} \end{bmatrix} \text{ and } \mathcal{A}_b = \begin{bmatrix} \frac{\dot{\xi}_1\mu_1\varpi_1}{\omega_1^2} \\ \frac{\dot{\xi}_2\mu_2\varpi_2}{\omega_2^2} \\ \frac{\dot{\xi}_3\mu_3\varpi_3}{\omega_3^2} \end{bmatrix}.$$

$\xi_i$, $\nu_i$ and $\omega_i$ are all readily computable variables as

$$\mu_i = (1 + \eta_i^2), \tag{14}$$

$$\omega_i = \xi_i(1 + \eta_i^2)(1 - \eta_i^2), \tag{15}$$

$$\nu_i = \frac{2\xi_i}{(\exp(\frac{e_{ci}}{2}) + \exp(-\frac{e_{ci}}{2}))^2}. \tag{16}$$

## III. MAIN RESULTS

### A. Optimal Controller Design

The performance index function is defined as follows:

$$\mathcal{V}(\varepsilon(t), u(t), t) = \int_t^\infty e^{-\rho(\tau-t)}(\mathcal{R}(\varepsilon(t), u(t))d\tau. \tag{17}$$

where $\mathcal{R}(\varepsilon(t), u(t)) = \varepsilon(t)^T\Omega\varepsilon(t) + u(t)^Tu(t)$, $\rho > 0$ represents the discount factor, and $\Omega$ is a positive defined matrix.

Then, the optimal value function is given by

$$\mathcal{V}^*(\varepsilon(t), u^*(t), t) = \min_{u(t)} \mathcal{V}(\varepsilon(t), u(t), t) \tag{18}$$

where $u^*(t)$ is the optimal controller. For simplicity, $\mathcal{V}^*(\varepsilon(t), u^*(t), t)$ is written as $\mathcal{V}^*$.

Based on Bellman's optimality principle, the following HJB equation is derived:

$$\mathcal{H}(\varepsilon(t), u^*(t), \mathcal{V}_\varepsilon^*, t) = \mathcal{V}_\varepsilon^* \big(\mathcal{A}_a(\dot{\mathbf{X}}(t) - \dot{\mathbf{X}}_d(t)) - \mathcal{A}_b\big) - \rho\mathcal{V}^*$$
$$+ \varepsilon(t)^T\Omega\varepsilon(t) + u(t)^T u(t)$$
$$= 0. \quad (19)$$

where $\mathcal{V}_\varepsilon^* = \partial\mathcal{V}^*/\partial\varepsilon$.

By using the stationarity condition $\partial\mathcal{H}(\varepsilon(t), u^*(t), \mathcal{V}_\varepsilon^*, t)/\partial u(t) = 0$, we have the optimal controller, formulated as

$$\mathbf{u}_i^* = -\frac{1}{2}g^T\mathcal{A}_a^T\mathcal{V}_\varepsilon^*. \quad (20)$$

By inserting (20) into (19), we find that the HJB equation is able to be rewritten as

$$\mathcal{V}_\varepsilon^*\big(\mathcal{A}_a(f(\mathbf{X}(t)) + d(t) - \dot{\mathbf{X}}_d(t)) - \mathcal{A}_b\big) - \rho\mathcal{V}^*$$
$$+ \varepsilon(t)^T\Omega\varepsilon(t) - \frac{1}{4}\mathcal{V}_\varepsilon^{*T}g\mathcal{A}_a\mathcal{A}_a^T g^T\mathcal{V}_\varepsilon^*$$
$$= 0. \quad (21)$$

In order to obtain the optimal controller of PMSM system, it is necessary to obtain the gradient term a of the optimal performance index, which is expected to be obtained by solving the HJB equation (21). However, due to unknown disturbances and inherent nonlinearity, it is impossible to solve the HJB equation by analytical method. Therefore, an ADP method combining historical data is proposed to solve the HJB equation (21) in the next section.

*B. Solutions of the HJB Equation*

An NN is introduced to approximate

$$\mathcal{V}^* = \mathbb{W}_{ci}^{*T}\sigma_{ci} + \varepsilon_{ci}, \quad (22)$$

where $\mathbb{W}_{ci} \in \mathbf{R}^{\tilde{n}c}$ denotes the weight vector, $\sigma_{ci} \in \mathbf{R}^{\tilde{n}c}$ is the basic function vector, $\tilde{n}c$ represents the number of neurons, and $\varepsilon_{ci}$ is a constant that denotes the approximation error.

Then, the estimated $\hat{\mathbf{u}}_i$ of optimal controller is obtained as

$$\hat{\mathbf{u}}_i = -\frac{1}{2}g^T\mathcal{A}_a^T\hat{\mathbb{W}}_{ci}^T\sigma_{ci}. \quad (23)$$

Substituting (22) and (23) into the Bellman residual error $e_{\mathcal{B}i}$ yields

$$e_{\mathcal{B}i} = \hat{H}_i(\underline{\epsilon}_i, \hat{\mathcal{J}}_{\underline{\epsilon}_i}, \hat{\mathbf{u}}_i, \hat{\omega}_i) - H_i^*(\underline{\epsilon}_i, \mathcal{J}_i^*, \mathbf{u}_i^*, \omega_i^*).$$

Then we define the term $\phi_i = \nabla\sigma_{ci}\big(\mathcal{A}_a(f(\mathbf{X}(t)) + d(t) - \dot{\mathbf{X}}_d(t)) - \mathcal{A}_b\big)$.

To mark sure that the Bellman residual error $e_{\mathcal{B}i}$ decays to 0, the gradient descent method is generally applied to the function $\frac{1}{2}e_{\mathcal{B}i}^T e_{\mathcal{B}i}$ by adjusting the updating law of the weight matrix $\hat{\mathbb{W}}_{ci}$. Historical state data is introduced into the following objective function in order to avoid PE conditions

$$\underline{E} = \frac{\frac{1}{2}e_{\mathcal{B}i}^T e_{\mathcal{B}i}}{(1 + \phi_i^T\phi_i)^2} + \sum_{p=1}^{\mathcal{N}_o}\frac{\frac{1}{2}e_{\mathcal{B}pi}^T e_{\mathcal{B}pi}}{(1 + \phi_{pi}^T\phi_{pi})^2}, \quad (24)$$

where $\phi_{pi}$ and $e_{\mathcal{B}pi}$ for $p = 1, 2, \ldots \mathcal{N}_o$ (note: $\mathcal{N}_o \leq \tilde{n}c$, $\tilde{n}c$ stands for the number of neurons used in the NN) are defined as below:

$$\phi_{pi} = \nabla\sigma_{ci}\big(\mathcal{A}_a(f(\mathbf{X}(t_p)) + d(t_p) - \dot{\mathbf{X}}_d(t_p)) - \mathcal{A}_b\big)$$
$$e_{\mathcal{B}pi} = \hat{\mathbb{W}}_{ci}^T\phi_{pi} + \varepsilon(t_p)^T\Omega\varepsilon(t_p) + \mathbf{u}(t_p)^T\mathbf{u}(t_p).$$

The gradient descent method is applied to the objective function $\underline{E}$. The updating law of $\hat{\mathbb{W}}_{ci}$ is obtained as follow with $t_p \in [t_j, t_{j+1}]$ denoting the index to mark the historical state time

$$\dot{\hat{\mathbb{W}}}_{ci} = -l_{ci}\frac{\partial\underline{E}}{\partial\hat{\mathbb{W}}_{ci}}$$
$$= -l_{ci}\frac{\phi_i e_{\mathcal{B}i}}{(1 + \phi_i^T\phi_i)^2} - \sum_{p=1}^{\mathcal{N}_o}l_{ci}\frac{\phi_{pi}e_{\mathcal{B}pi}}{(1 + \phi_{pi}^T\phi_{pi})^2}, \quad (25)$$

where $(1 + \phi_i^T\phi_i)^{-2}$ and $(1 + \phi_{pi}^T\phi_{pi})^{-2}$ are the normalization terms and $l_{ci} > 0$ is the design parameter.

Define the weight estimation error as $\tilde{\mathbb{W}}_{ci} = \mathbb{W}_{ci}^* - \hat{\mathbb{W}}_{ci}$, then based on (26), one has

$$\dot{\tilde{\mathbb{W}}}_{ci} = -l_{ci}(\varphi_i\varphi_i^T + \sum_{p=1}^{\mathcal{N}_o}\varphi_{pi}\varphi_{pi}^T) + \frac{l_{ci}\varepsilon_{Hi}}{(1 + \phi_i^T\phi_i)^2}$$
$$+ \sum_{p=1}^{\mathcal{N}_o}\frac{l_{ci}\varepsilon_{Hpi}}{(1 + \phi_{pi}^T\phi_{pi})^2}, \quad (26)$$

where
$$\varphi_i = \frac{\phi_i}{(1 + \phi_i^T\phi_i)^2}, \qquad \varphi_{pi} = \frac{\phi_{pi}}{(1 + \phi_{pi}^T\phi_{pi})^2},$$
$$\varepsilon_{Hi} = \nabla\varepsilon_{ci}\big(\mathcal{A}_a(f(\mathbf{X}(t_p)) + d(t_p) - \dot{\mathbf{X}}_d(t_p)) - \mathcal{A}_b\big),$$
$$\varepsilon_{Hpi} = \nabla\varepsilon_{ci}\big(\mathcal{A}_a(f(\mathbf{X}(t)) + d(t) - \dot{\mathbf{X}}_d(t)) - \mathcal{A}_b\big).$$

## IV. SIMULATION VERIFICATION

In order to verify the proposed control method, a simulation example based on PMSM is implemented. Figure .1 shows the PMSM system status tracking diagram. It can be seen that the control method proposed in this paper can effectively make the PMSM system state track the reference trajectory. Figure .2 shows the tracking error diagram. Figure .2 shows that all errors are limited to a given prescribed performance boundary

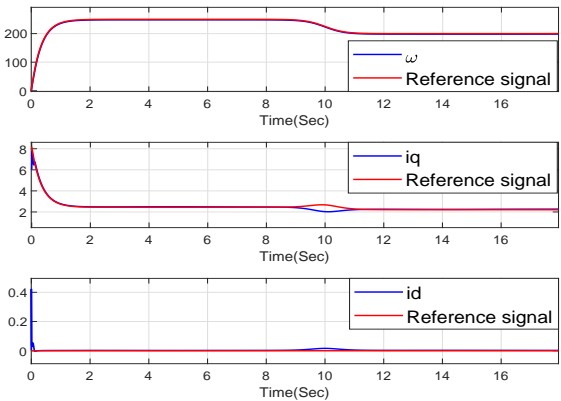

Fig. 1. State trajectories of the PMSM system

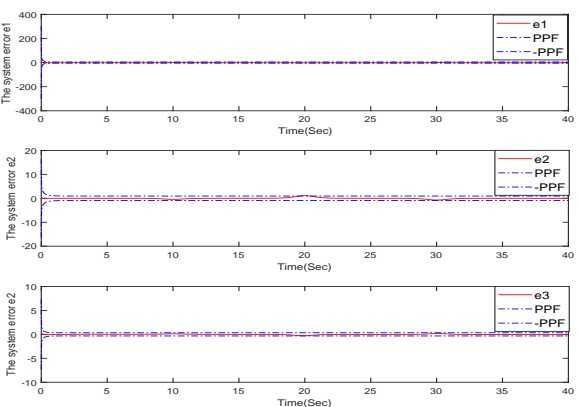

Fig. 2. State trajectories of the PMSM system

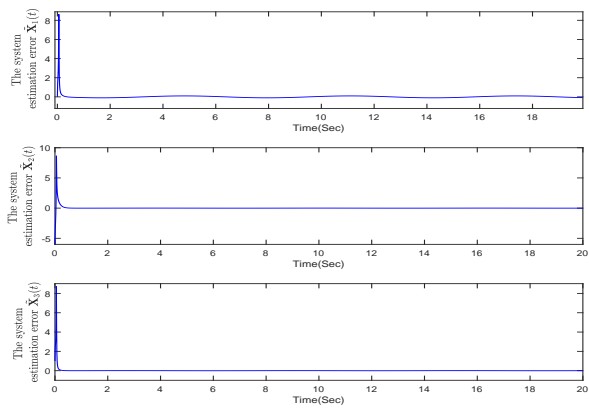

Fig. 3. State trajectories of the PMSM system

## V. CONCLUSION

This paper have proposed a prescribed performance optimal control method for permanent magnet synchronous motor (PMSM) systems with nolinear disturbance. Firstly, a novel prescribed performance approach has been proposed to constrain both the state and error of PMSM systems in order to obtain high precision control performance. Then, an adaptive dynamic programming method based on historical data has been designed to solve the optimal controller. Furthermore, an identifier based on neural network has been constructed to approximate the system disturbances. Finally, a simulation example has been implemented to validate the efficacy of the proposed control method.

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
