# OpenReview forum: "Historical Data-Based Prescribed Performance Optimal Control for PMSM Systems with Disturbance"
_IEEE.org/ICIST/2024/Conference — IEEE ICIST 2024 Conference Submission_

### Official Review · Reviewer_TbY4 · 2024-08-20
**Lack of innovation in research**

**Rating:** 3
**Confidence:** 5

**Review:**

1. The current work does not demonstrate sufficient novelty or advancement over existing literature to merit publication.
2. The writing quality falls short of the standards. The reviewers suggest significant revisions to improve clarity and conciseness.

---

### Official Review · Reviewer_2kzQ · 2024-08-21
**Lack of innovation**

**Rating:** 3
**Confidence:** 5

**Review:**

1. The main controbation of this paper is not clear.
2. The literature review in the Introduction surveys some previous work on the solutions and methods to address prescribed performance
optimal control, while comparation among these methods which is the most important part is neglected.

---

### Official Review · Reviewer_4RKH · 2024-08-24
**Review Comments for Manuscript No. 194**

**Rating:** 6
**Confidence:** 4

**Review:**

1.The manuscript lacks proper logical transitions between sections. The narrative should be improved to ensure that each section flows naturally into the next.

2.There are noticeable errors in the references, which need to be corrected. Additionally, there are several apparent typographical or formatting mistakes within the manuscript. The authors should thoroughly proofread the manuscript and make the necessary corrections.

3.The manuscript focuses on the Historical Data-Based Method and PP-based PMSM control. The authors should address the following points:

3.1. Is the Historical Data-Based Method an original contribution by the authors? The manuscript mentions that historical state data is introduced to avoid persistent excitation (PE) conditions. However, it should clarify whether the information provided by historical data is equivalent to or surpasses the information gained through PE. Does this method impose any specific requirements on the duration of data collection?

3.2. Apart from the Historical Data-Based Method, how does the theoretical derivation in the remaining parts of the manuscript compare to and relate to the backstepping-based nonlinear control method? The authors should clarify the distinctions and connections between these approaches.

4.The titles of all the figures in the simulation section are identical, which reduces clarity. Each figure should have a distinct and descriptive title. Additionally, Figure 3 is not cited in the manuscript and should be properly referenced.

---

### Decision · Program_Chairs · 2024-09-08

Reject